# The Application of Nano-MoS_2_ Quantum Dots as Liquid Lubricant Additive for Tribological Behavior Improvement

**DOI:** 10.3390/nano10020200

**Published:** 2020-01-23

**Authors:** Junde Guo, Runling Peng, Hang Du, Yunbo Shen, Yue Li, Jianhui Li, Guangneng Dong

**Affiliations:** 1School of Mechatronic Engineering, Xi’an Technological University, Xi’an 710021, China; pengrunling@163.com (R.P.); shenyunbo@xatu.edu.cn (Y.S.); 2Institute of Machinery Manufacturing Technology, China Association of Employment Promotion, Mianyang 621900, China; 3School of Science, Xi’an Jiaotong University, Xi’an 710049, China; 4Key Laboratory of Education Ministry for Modern Design and Rotor-Bearing System, Xi’an Jiaotong University, Xi’an 710049, China

**Keywords:** MoS_2_ QDs, lubricant additive, tribological property, lubrication mechanism

## Abstract

Molybdenum disulfide quantum dots (MoS_2_ QDs) are a promising lubricant additive for enhanced engine efficiency. In this study, MoS_2_ QDs were used as lubricating oil additives for ball-on-disc contact and had adequate dispersity in paroline oil, due to their super small particle size (~3 nm). Tribological results indicate that the friction coefficient of paroline oil with 0.3 wt.% MoS_2_ QDs reached 0.061, much lower than that of pure paroline oil (0.169), which is due to the formation of a stable tribo-film formed by the MoS_2_, MoO_3_, FeS, and FeSO_4_ composite within the wear track. Synergistic lubrication effects of the tribo-film and ball-bearing effect cooperatively resulted in the lowest friction and wear.

## 1. Introduction

Friction and wear between moving components have an important influence on the energy loss of engineering equipment [1,2,3]. Lubrication technology is of great significance to improve the tribological properties and reduce the wear of friction pairs [4,5]. In addition, lubricants have the functions of cleaning wear debris, sealing clearances, dissipating heat, and resisting corrosion or rust [6,7]. Lubricant additives can reduce friction loss and conservation resources [8]. Lubricant additives for lubricating oil are widely used to improve durability of frictional components [9]. In recent years, the tribological property of nanoparticle additives attracted a great deal of attention, as they show friction-reducing and anti-wear effects to a certain level. However, the production of the above additives creates a potential environmental problem due to hazardous emissions such as H_2_S and phosphorus, as well as sludge obtained during purification. Recent environmentally friendly regulations advocate for low-phosphorus or phosphorus-free industrial lubricants [10]. Lubricant additives were extensively studied in the past few years, such as MoS_2_ nanosheets [11], MoS_2_ nanotubes [12], fullerene-like MoS_2_ [13], and grapheme [14].

However, the limitation of dispersibility is the key factor affecting the tribological properties of lubricant additives in liquid, and the particle size and thickness of lubricant additives have a great influence on their tribological properties [15]. The dispersibility of the nano-additive in lubricating oil shows excellent performance; thus, MoS_2_ quantum dots (QDs) show great potential in lubricating oil additives based on tribological properties.

Nanometer-level additives exhibit excellent performance in terms of dispersibility in lubrication oil; thus, MoS_2_ QDs show great potential as lubricant additives in terms of tribological performance [16]. Fullerene-like nano-MoS_2_ particles can easily penetrate the contact area and form homogeneous tribo-film to achieve low friction and wear [13]. There were many studies on the fabrication of nano-MoS_2_ as oil additives [11,17,18]; however, it is very difficult to synthesize MoS_2_ quantum dots with particle size less than 5 nm as lubricant additives compared to MoS_2_ nanosheets. Thus, MoS_2_ QDs have a smaller particle size and higher special surface area, which is more beneficial to dispersibility, resulting in them having great potential to be used as lubricant additives [19]. Moreover, the tribological performance of MoS_2_ nanosheet and nanotube lubricant additives was studied, but the tribological behavior of MoS_2_ QDs (size < 5 nm) is still unclear.

This study aims to investigate the tribological performance of MoS_2_ QDs as a lubricant additive in paroline oil. Ball-on-disc tribological tests were conducted under boundary lubrication to simulate the point contact between mechanical components. The influence of the additive amount of MoS_2_ QDs on the tribological performance was investigated. In order to clearly investigate the lubrication mechanism of MoS_2_ QDs in paroline oil, and to differentiate the competitive adsorption from other lubricant additives (dispersant or detergent), MoS_2_ QDs were used as the single lubricant additive in paroline oil, thereby obtaining sufficient tribological results in laboratory tests.

## 2. Experimental

### 2.1. Synthesis of MoS_2_ QDs and Dispersion Process in Paroline Oil

The MoS_2_ QDs were synthesized through a one-pot hydrothermal method with a mixture of Na_2_MoO_4_·2H_2_O and thiocarbamide. In a typical procedure, the mixture of Na_2_MoO_4_·2H_2_O (120 mg) and thiocarbamide (324 mg) was dispersed in 10 mL of distilled water. After stirring for 30 min, the mixture was transferred to a 50-mL poly (tetrafluoroethylene) (Teflon) autoclave and kept at 200 °C for 8 h through microwave irradiation in nitrogen atmosphere. After being cooled to room temperature naturally, the product of the reaction was repetitively purified by centrifuging in ethanol and deionized water. Furthermore, the precipitate obtained by centrifugation was added to 10 mL of *N*-methylpyrrolidone, which was continuously smashed by a wheel sonde for 16 h. Then, the product was separated centrifugally at 5000 rpm for 30 min. The supernatant MoS_2_ QDs represented the target dispersion solution. Furthermore, MoS_2_ QDs could be obtained via a filtration or centrifugation method.

The final prepared MoS_2_ QDs were a black powder, which were added to paraffin oil by ultrasonic dispersion to achieve stable dispersions.

### 2.2. Tribological Procedure of Ball-On-Disc Testing

The tribological property was tested using a ball-on-disc tribometer (UMT-2, CETR Corporation Ltd., Campbell, USA) in a reciprocation model. Schematic diagram of ball-on-disc tribometer was shown in Figure 1. The experiments were performed for 30 min at room temperature (20–23 °C). All tests were repeated three times. The disc was bearing steel (surface roughness of about 0.02 μm), and the ball was industrial bearing steel (Φ9.5 mm, surface roughness 0.01 μm). The applied load of the ball-on-disc tribometer was 6 N, the applied sliding speed was 1.5 Hz, and the sliding travel was 6mm. The above testing condition was selected based on some types of gears with low speed and moderate-duty gears.

### 2.3. Characterization

X-ray diffraction (XRD) results were obtained on a D8 Advance X-ray diffractometer (D8 advance, Bruker, Karlsruhe, Germany). X-ray photoelectron spectroscopy (XPS, AXIS Ultrabld, Kratos, UK) was used to analyze the structure of MoS_2_ QDs. Transmission Electron Microscope (TEM, Tecna^TM^ G^2^F30, FEI Company, Hillsboro, OR, USA) was used to observe the morphology of MoS_2_ QDs. The viscosity of lubricating oils was tested using a capillary viscometer. The wear scars of GCr15 balls were observed by an optical microscope. The wear scars and Energy Dispersive Spectrum (EDS, SU3500, Oxford Instruments, Abingdon, Oxfordshire, UK) images of GCr15 discs were obtained by scanning electron microscopy (SEM, VEGA 3 LMH/LMU, TESCAN, Brno, Czech Republic), with an accelerating voltage of about 20 kV and a corresponding current value of 7 μA. Three-dimensional (3D) profiles of the worn surface were obtained using a laser scanning confocal microscope (OLS4000, OLYMPUS Company, Tokyo, Japan). 

## 3. Results and Discussion

### 3.1. Characterization of MoS_2_ QDs

TEM images of the resultant MoS_2_ QDs are shown in Figure 2. The average diameter of MoS_2_ QDs was about 3 nm, indicating the excellent monodispersity of MoS_2_ QDs, which were well dispersed and ranged from 1.5 to 5 nm (Figure 2a). The paralleled and ordered lattice fringe can be observed in the TEM map shown in Figure 2b, which illustrates the high crystallinity of the MoS_2_ QDs. The lattice fringe spacing was about 0.2 nm, which matches well with previously reported values for crystal MoS_2_ [20]. At this size, MoS_2_ QDs are small enough to have quantum confinement effects and small-size effects. X-ray diffraction (XRD) spectroscopy was used to test the crystal structure of MoS_2_ QDs. As shown in Figure 2c, bulk MoS_2_ (before being smashed by the wheel sonde) had several obviously strong diffraction peaks at 2θ = 32.7°, 39.6°, 49.8°, 58.3°, and 60.4°, which were assigned to the (100), (103), (105), (110), and (112) faces, respectively, indicating that bulk MoS_2_ has a multilayer structure. Moreover, some lower peaks at 2θ = 29°, 2θ = 39.6°, 2θ = 44.2°, 2θ = 49.8°, and 2θ = 60.2° could be observed, which were assigned to the (004), (103), (006), (105), and (008) faces, respectively (JC-PDF (03-065-1951)). In addition, no peaks could be observed in the XRD patterns of MoS_2_ QDs, since the materials were thin or presented as monolayers, revealing no intense interference on the aligned crystal planes [21]. The XRD result indicated that the resultant product was a thin layer.

The photoluminescence (PL) spectra of the MoS_2_ QDs aqueous solutions were measured at various excitation wavelengths, as shown in Figure 2d. With excitation wavelengths increasing from 290 to 370 nm, the PL emission peaks shifted to longer wavelengths. The PL spectra of the MoS_2_ QD suspension exhibited a strong emission peak at 430 nm under an excitation wavelength of 360 nm. With the increase in excitation wavelength from 360 to 450 nm, the PL emission peaks shifted from 430 to 530 nm. These are typical size-dependent PL properties [22]. The property of excitation-dependent PL indicated the polydispersity of the as-prepared MoS_2_ QDs [23].

### 3.2. Friction Property of MoS_2_ QDs in Paroline Oil

The tribological performance of paraffin oils with different MoS_2_ QD loading as a lubricant additive is shown in Figure 3. The coefficients of friction (COF) of paraffin oils with different additive amounts of MoS_2_ QDs were measured at an applied load of 6 N and reciprocating frequency of 1.5 Hz, as presented in Figure 3a,b. One can observe relatively larger fluctuations in the friction response of pure paraffin oils, in comparison to the pure paraffin oil samples with MoS_2_ QDs, whose COF was as high as 0.169, and the COF was found to increase with time in the initial stage during the running-in period.

Along with the loading of MoS_2_ QDs into paraffin oil, the COF of the nanoparticle oil reduced by much more than that of pure paraffin oil. The lowest COF of 0.061 was obtained by the oil sample with 0.3 wt.% MoS_2_ QDs. Moreover, Figure 3b (partial enlargement of Figure 3a) shows the influence of particle concentration on the COF of the MoS_2_ QD oil suspension, indicating that the average COF was influenced by the MoS_2_ QD concentration. The average COF obviously fell from 0.16 to 0.061 in the range of 0.1 to 0.5 wt.% MoS_2_ QDs, reflecting that the addition of nanoparticle lubricants strengthened the sliding response when stabilizing for additive amounts below 0.3 wt.%. However, the average COF showed a slightly increasing trend from 0.061 to 0.065 in the range of 0.3 wt.% to 0.5 wt.% MoS_2_ QDs. Furthermore, the COF tended to be stable after rubbing, and the corresponding average COF in the stable period for higher concentrations of MoS_2_ QDs presented a lower anti-frictional property, as shown in Figure 3c, which is consistent with our previous report [24]. The relevant tribological mechanism is due to the MoS_2_ QDs particles at higher concentration accumulating in the inlet of the ball-on-disc contact area, which causes an insufficient supply of lubricant and starvation in the contact zone [25]. The running-in period is of great significance to the regulation of tribological performance to a certain extent. Reducing the running-in period is beneficial to improving the anti-frictional property. The formation of a boundary lubrication film is the main reason for the stability of the friction coefficient. Figure 3d shows the variation of running-in time with the addition of MoS_2_ QDs. The rubbing period of nano-oil with 0.1 wt.% concentration lasted longer than the others, with a time of 460 s. In addition, it is noteworthy that the rubbing time obviously decreased with the increase in concentration. Although the 0.5 wt.% sample had the shortest rubbing time in terms of friction property, the corresponding average COF was slightly higher than that of the 0.3 wt.% oil sample under the same conditions. 

### 3.3. Wear Scar Images of Balls and Discs

Figure 4 reveals the wear scar maps of the tested balls sliding against the bearing steel discs. It shows that the wear degree of the samples containing MoS2 QDs was obviously improved, compared to that of the paroline base oil. When the base oil was a lubricant, the worn surface of the ball presented a black sediment, which confirmed that the main component was Fe, resulting from ball-on-disc milling, as shown in Figure 4b (EDS). Moreover, the 0.1 wt.% and 0.2 wt.% samples also presented black sediment, but the wear radiuses were visibly reduced. In particular, the radius of 0.3 wt.% MoS_2_ QDs was smaller than that of the other samples, showing the superior anti-wear performance of MoS_2_ QDs at this specific percentage. Although the 0.5 wt.% oil sample had a lower COF value in the stabilized state after rubbing, the corresponding steel ball sliding with the disc had a higher wear rate.

To further investigate the wear mechanism, the SEM images of worn surfaces of the lower disc samples lubricated by pure paroline oil and the paroline oil samples containing 0.1–0.5 wt.% MoS_2_ QDs were investigated (Figure 5). It can be seen from Figure 5a,b that there was obvious ploughing and some pits on the worn track, which were due to the local second rupture of debris during the sliding process; moreover, the friction of the debris at the sliding interface led to clear furrows on the worn surface. Figure 5c,d show that the worn surface presented finer mesh-like grooves, in agreement with the COF results in Figure 3. For the 0.3 wt.% sample, the worn surface was covered by fine grooves and detachments after the sliding test, as shown in Figure 5e,f, and the wear mechanism was dominated by microploughing. Furthermore, it can be seen from Figure 5g that the wear scratch was almost invisible; following the local zoom in Figure 5f, only slight furrows could be observed. This is mostly attributed to the increase in MoS_2_ QDs, which could reduce the wear of the frictional interfaces. This indicates that MoS_2_ QDs played a lubricating role and prevented the ploughing wear in the sliding process. The above results are in agreement with the tribological result showing the decrease in friction coefficient in Figure 3.

The 3D profiles of the worn surfaces for different lubricant oils after the wear tests are shown in Figure 6. Compared with pure paroline oil, MoS_2_ QDs had a significant improvement in the wear surface. After the wear test at room temperature, only small grooves could be observed, and the wear mechanism was mainly formed by microploughing. The 3D maps of the wear scars of pure paroline oil are shown in Figure 6a, showing the obvious furrow and indentation, and the corresponding maximum depth was 2.8 μm. Although the wear depth for the 0.1 wt.% sample was much smoother than that of pure paroline oil, microploughing still existed, with a corresponding wear depth of 0.09 μm. Moreover, the anti-wear property improved with the amount of MoS_2_ QDs. This indicates that MoS_2_ QDs in paroline oil prevented the ploughing wear that was evident in the control sample.

### 3.4. Lubrication Mechanism

Figure 7 reveals the presence of Mo and S elements within the wear track when lubricated with paroline and 0.2 wt.% MoS_2_ QDs. As shown in Figure 7d,e, the EDS mapping results clearly confirmed the presence of Mo and S elements within the wear track, suggesting that the capping layer of MoS_2_ QDs also contributed to forming the tribo-film. The atomic ratio of elements Mo and S was 2.17:1, as shown in Figure 7c, suggesting the formation of a tribo-film composed of Mo and S.

XPS was employed to study the surface chemical composition of the wear scars of the steel discs, which were lubricated by paroline oil with MoS_2_ QDs, revealing visible Mo and S peaks. As shown in Figure 8, the XPS peaks of Mo in Figure 8a were mainly attributed to Mo–O or Mo–S. The Mo and the S peaks at 235.7 eV and 165.1 eV corresponded to MoS_2_, confirming the formation of the MoS2 tribo-film on the wear scar, in agreement with the O–S of O1*s* at 531.6 eV in Figure 8b,c [26]. The peak of the bond energy of 168.9 eV in Figure 8b was attributed to S2*p*, i.e., the bond of sulfur to oxygen (S–O).

The peak in bonding energy at 168.9 eV was attributed to S2*p* in Figure 8b, i.e., the bonding of sulfur to oxygen (S–O) [27]. Accordingly, it shows the presence of oxides of Mo in the tribo-film, whereas the peak at 226.3 eV was attributed to S (S2*s*) rather than Mo [13]. The peak of S2*p* appeared at the binding energy of 169.3 eV, corresponding to the S^4+^ state realized in FeSO_4_ [28]. The peak component of S2*p* at the low bonding energy of 162.1 eV was attributed to the presence of Fe-S [13]. Thus, the above results reveal that fewer nano-MoS_2_ QDs were oxidized to MoO_3_, and the tribo-film with MoS_2_ QDs had more MoS_2_ but less MoO_3_ [29]. These peaks demonstrate that the patched composite film consisted of MoS_2_, FeS, and sulfate, which was the source of improved tribological performance and enhanced lubricant durability. The combination of MoS_2_ QDs and paroline oil facilitated the formation of the tribo-film.

According to the above results, a ball-on disc schematic diagram of the sliding process is shown in Figure 9. Paroline oil can adsorb onto the rubbing interface and form a tribo-film, which plays a protective role during the friction process. With the addition of MoS_2_ QDs, the nano-additives and paroline oil self-assemble and form a nanocomposite layer on the sliding regions during the rubbing process, such as MoS_2_, MoO_3_, FeS, FeSO_4_, etc., thus minimizing friction and wear. The improvement in tribological performance indicates that the composite oil can easily form a tribo-film and adsorb onto the wear track during the sliding process. Therefore, MoS_2_ QDs have anti-wear and anti-friction properties as a lubricating oil additive. In addition, the amount of MoS_2_ QDs plays a decisive role in tribological performance. With the increase in additive, the viscosity of the composite lubricant oil exhibited an increasing trend, but the change was not the main factor behind the boundary lubrication regime for the ball-on-disc contact. MoS_2_ QDs can be sustainably dispersed in paraffin oil. In the 10-day dispersion experiment, the oil samples containing MoS_2_ QDs had good dispersion, and no particle sedimentation occurred. 

Because the spherical MoS_2_ QDs function as ball-bearing lubricants in the process of friction, they can be adsorbed onto the slide track by adding MoS_2_ QDs to pure paroline oil to prevent the frictional component from contacting it directly [30,31]. In summary, the synergistic lubrication effect of the tribo-film, the improvement of the bearing capacity, and the ball-bearing effect together lead to the lowest friction and wear.

## 4. Conclusions

1. Monodisperse MoS_2_ QDs ranging from 1.5 to 5 nm were prepared, with a paralleled and ordered lattice fringe, and a lattice fringe spacing of about 0.2 nm. In addition, MoS_2_ QDs were sustainably dispersed in paraffin oil, showing no particle sedimentation in the 10-day dispersion experiment.

2. With the addition of MoS_2_ QDs to the paraffin oil, the lowest COF of 0.061 could be obtained by the 0.3 wt.% MoS_2_ QD oil sample, which dropped by approximatively 64% compared to the pure paraffin base oil. Furthermore, the rubbing time obviously decreased with the increase in MoS_2_ QDs.

3. The main wear type of the worn surfaces lubricated by the paroline base oil with MoS_2_ QDs could be attributed to slight ploughing wear. Pure paroline oil presented an obvious furrow and indentation, with a corresponding maximum depth of 2.8 μm.

4. The MoS_2_ QDs play a decisive role in the improvement of the tribological performance. The potential lubrication mechanisms include the formation of a composite tribo-film composed of MoS_2_, MoO_3_, FeS, and FeSO_4_. Additionally, spherical MoS_2_ QDs have aprobable ball-bearing lubrication effect during the friction process.

## Figures and Tables

**Figure 1 nanomaterials-10-00200-f001:**
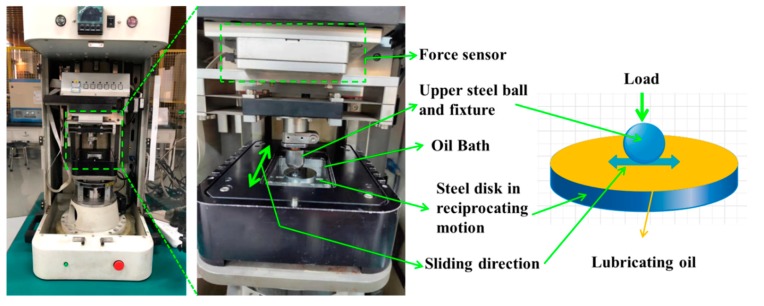
Schematic diagram of ball-on-disc tribometer.

**Figure 2 nanomaterials-10-00200-f002:**
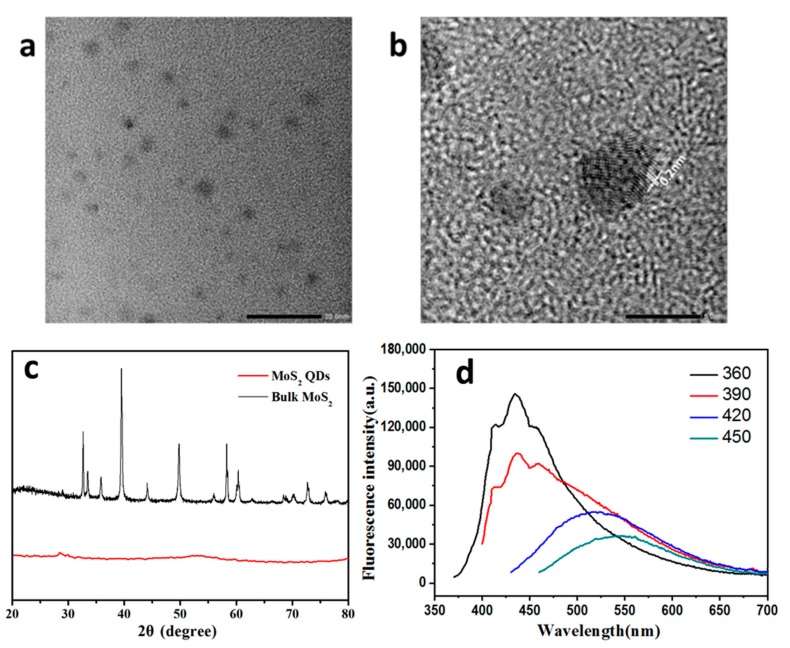
(**a**) TEM pattern of MoS_2_ quantum dots (QDs); (**b**) partial enlarged drawing of (**a**); (**c**) XRD pattern of MoS_2_ QDs; (**d**) fluorescence spectrum of MoS_2_ QDs.

**Figure 3 nanomaterials-10-00200-f003:**
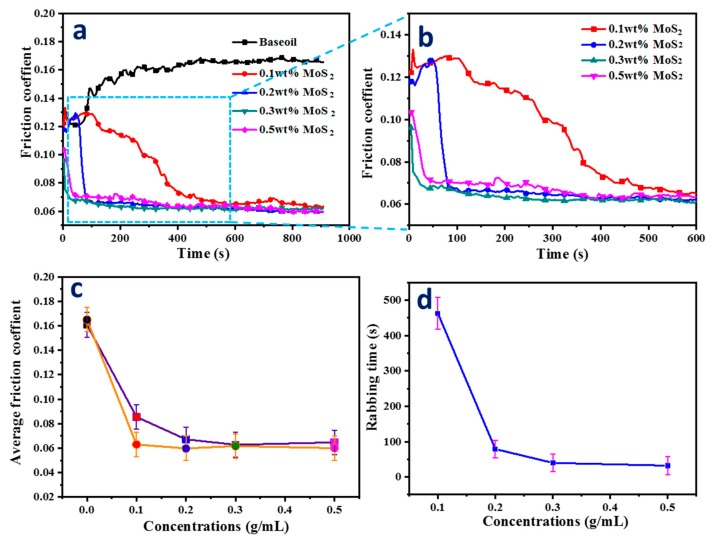
(**a**) Tribological tests results of paraffin oils with different additive amounts of MoS_2_ QDs; (**b**) coefficient of friction (COF) for different additive amounts of MoS_2_ QDs; (**c**) average COF and COF in steady state after rubbing; (**d**) the change in rubbing time at different concentrations.

**Figure 4 nanomaterials-10-00200-f004:**
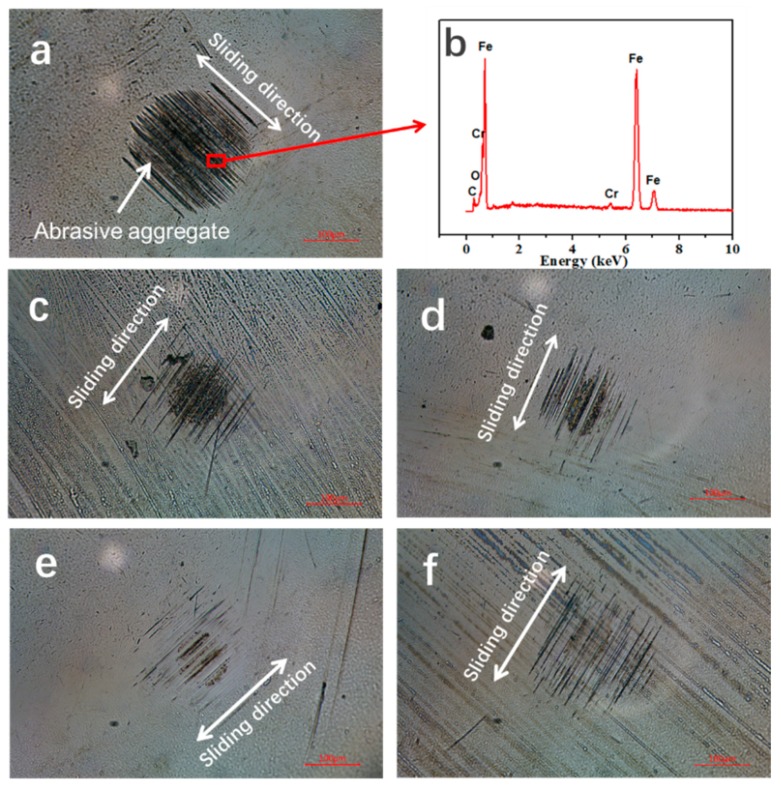
(**a**) The ball wear scar map pure base oil; (**b**) EDS pattern of pure base oil; (**c**) 0.1 wt.% MoS_2_ QDs; (**d**) 0.2 wt.% MoS_2_ QDs; (**e**) 0.3 wt.% MoS_2_ QDs; (**f**) 0.5 wt.% MoS_2_ QDs.

**Figure 5 nanomaterials-10-00200-f005:**
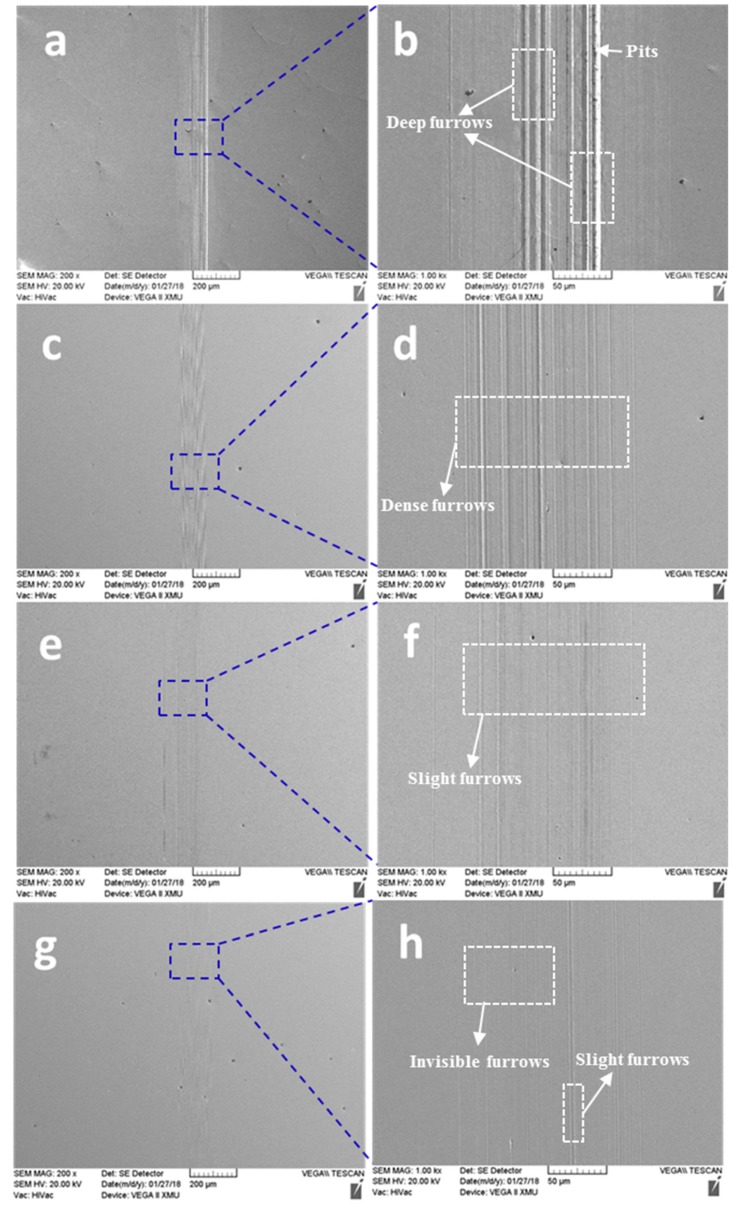
SEM wear images of wear scars after tests using different concentrations of paroline oils: (**a**) pure paroline oil; (**b**) paroline oil with 0.1 wt.% QDs; (**e**) paroline oil with 0.3 wt.% QDs; (**g**) paroline oil with 0.5 wt.% MoS_2_ QDs. (**b**,**d**,**f**,**h**) are the local enlarged drawings of (**a**,**c**,**e**,**g**), respectively.

**Figure 6 nanomaterials-10-00200-f006:**
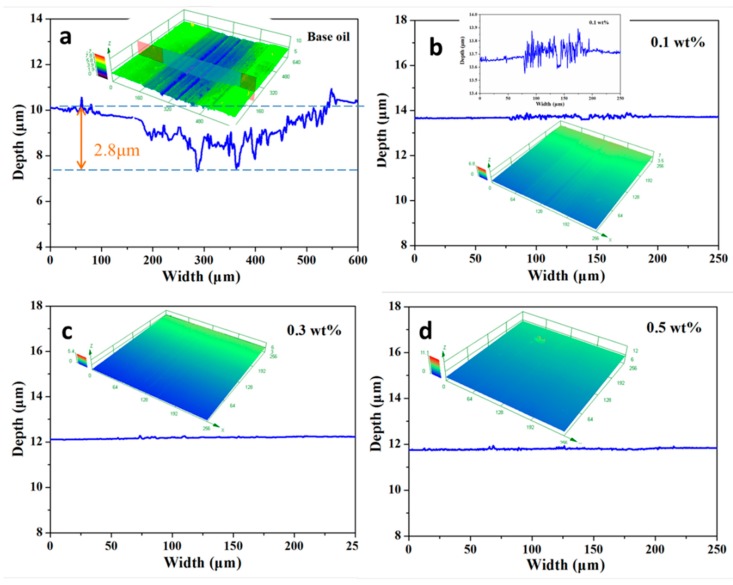
Three-dimensional (3D) wear profiles for different lubricants: (**a**) pure paroline oil; (**b**) paroline oil loading 0.1 wt.% MoS_2_ QDs; (**c**) paroline oil loading 0.3 wt.% MoS_2_ QDs; (**d**) paroline oil loading 0.5 wt.% MoS_2_ QDs.

**Figure 7 nanomaterials-10-00200-f007:**
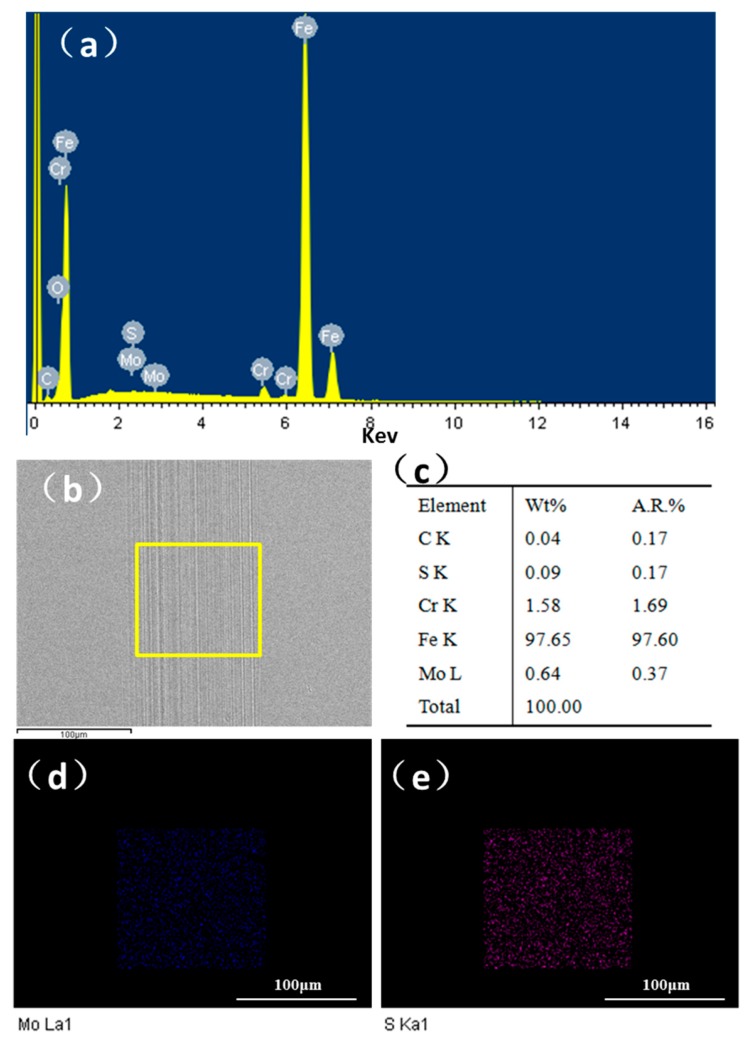
EDS characterization within the wear track: (**a**) EDS pattern of wear track for bottom disc; (**b**) SEM image of wear track; (**c**) EDS report of different elements; (**d**) elemental mapping characterization of Mo on the wear track; (**e**) elemental mapping characterization of S on the wear track.

**Figure 8 nanomaterials-10-00200-f008:**
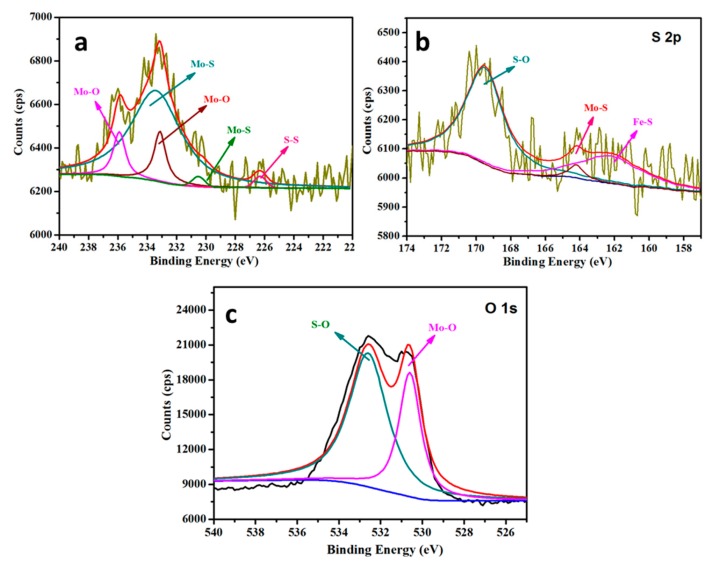
XPS results of wear scars on the steel discs lubricated by paroline oil with MoS_2_ QDs: (**a**) Mo3*d*; (**b**) S2*p*; (**c**) O1*s*.

**Figure 9 nanomaterials-10-00200-f009:**
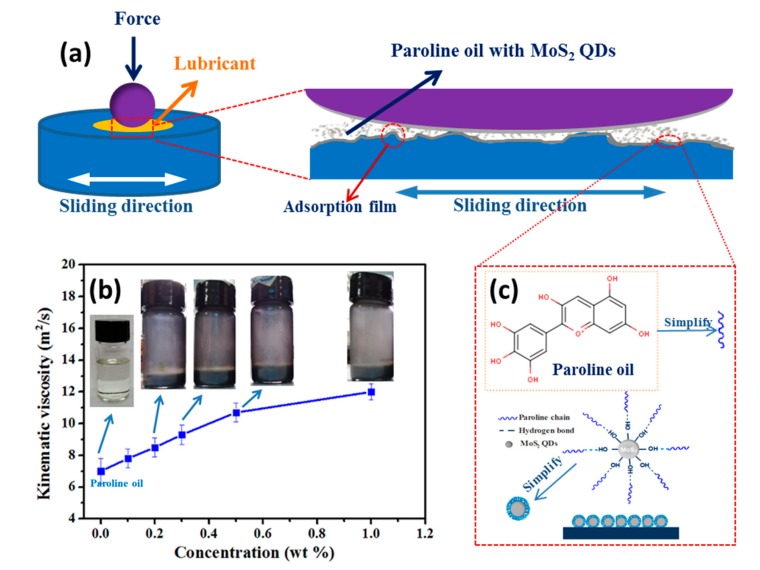
Schematic diagram of the lubricating mechanism: (**a**) ball-on-disc friction model; (**b**) viscosity with respect to additive amount; (**c**) adsorption process of paroline oil with MoS_2_ QDs.

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
