# Peer review of "The Application of Nano-MoS2 Quantum Dots as Liquid Lubricant Additive for Tribological Behavior Improvement"

_nanomaterials, 2020, doi:10.3390/nano10020200_

Round 1
Reviewer 1 Report
The subject matter is very interesting, important and has a special value considering practical applications. However, there are still some things that could be improved before publication. Therefore, I suggest a mandatory revision of the following points to increase the quality of the paper:
1) Why were load and sliding speed fixed at 6N and 1.5Hz?
2) The schematic diagram of ball-on-disk tribometer is not precise. There is no oil in figure 1. In my opinion Authors ought to show a picture of tribometr.
3) There no parameters of SEM for example accelerating voltage that influence on the EDS microanalysis resolution.
4) In Characterization Authors have written that they have used SEM, S-3000N, Hitachi. But in Fig.5 they have shown SEM wear images that were made with SEM Tescan Vega3.
5) Why there are no Mo and S in Fig.4b? It requires the comments.
6) Authors should show scales in Fig. 7a and 7b and SEM image of the wear track. Moreover errors are very big for molybdenum and sulfur (0,14% and 0,17%). It requires the comments.
7) How have the authors ensured continuous the paroline oil with MoS2 QDs access between samples?
8) In chapters 3.3 and 4 they have written “Formation of the composite tribo-film composed of MoS2, MoO3, FeS, FeSO4”. Do they have research that confirms this? Are these just supposition? It requires the comments.
Author Response
Comments: The subject matter is very interesting, important and has a special value considering practical applications. However, there are still some things that could be improved before publication. Therefore, I suggest a mandatory revision of the following points to increase the quality of the paper:
Answer: Thank you for your valuable comments. We are extremely thankful to these helpful comments on our manuscript. Based on these comments, we have answered the questions and revised them in detail one by one. 1) Why were load and sliding speed fixed at 6N and 1.5Hz?
Answer: Thank you very much for your comment.
The application of lubricating oil was widely used on different types of gears for engine. The load of 6N (about 580 MPa), 1.5Hz (18 mm/s) is selected based on one type of gears with a common working condition. In the early stage of the tribological test, we also tested the friction coefficients under different conditions. The largest distinction for tribological property can be reflected under this selected condition in the preliminary exploratory tests. Besides, we also refer to some similar testing method as shown in the following articles:
Wu H, Li X, He X, et al. An investigation on the lubrication mechanism of MoS2 nanoparticles in unidirectional and reciprocating sliding point contact: the flow pattern effect around contact area[J]. Tribology International, 2018, 122: 38-45. Wu H , Wang L , Johnson B , et al. Investigation on the lubrication advantages of MoS2, nanosheets compared with ZDDP using block-on-ring tests[J]. Wear, 2017:S004316481730830X.
2) The schematic diagram of ball-on-disk tribometer is not precise. There is no oil in figure 1. In my opinion Authors ought to show a picture of tribometr.
Answer: We are sorry for this imprecise expression. We have added the image of tribometer and revised the ball-on-disk schematic diagram in the revised manuscript as shown in figure 1.
3) There no parameters of SEM for example accelerating voltage that influence on the EDS microanalysis resolution.
Answer: Thank you for your careful reading of our manuscript. The relevant parameters have been added in Sec. 2.3. The accelerating voltagein the SEM was about 20 kV.
4) In Characterization Authors have written that they have used SEM, S-3000N, Hitachi. But in Fig.5 they have shown SEM wear images that were made with SEM Tescan Vega3.
Answer: We are sorry for this mistake. We have corrected them in Sec. 2.3 as marked in different color.
5) Why there are no Mo and S in Fig.4b? It requires the comments.
Answer: Thank you for your comment. Figure 4b is the EDS image for the pure paroline oil, we did not add MoS2 QDs in this oil sample, this is used to be a contrasted with nano-MoS2 oil samples.
6) Authors should show scales in Fig. 7a and 7b and SEM image of the wear track. Moreover errors are very big for molybdenum and sulfur (0,14% and 0,17%). It requires the comments.
Answer: Thank you for your careful comments. The scales have been added in figure 7, and the EDS measurements were retested as shown in Sec. 3.3. The new results is that the atomic ratio of elements Mo and S is 2.17:1, closed to 2:1.
7) How have the authors ensured continuous the paroline oil with MoS2 QDs access between samples?
Answer: Thank you for your careful and rigorous comment. In this article, the sliding samples were submerged in the lubricating oil, especially the contact area of ball-on-disk will be completely immerged in the lubricating oil. The supply of the lubricating oil is sufficient, and there is no starved oil-supply condition of lubricating oil. Besides, the starved condition is still a problem in the actual equipment operation for lubricating oils. The study is in the exploratory stage, and the relevant tests are completely tested in a laboratory environment. In the subsequent practical application, we will consider continuous the paroline oil with MoS2 QDs access.
8) In chapters 3.3 and 4 they have written “Formation of the composite tribo-film composed of MoS2, MoO3, FeS, FeSO4”. Do they have research that confirms this? Are these just supposition? It requires the comments.
Answer: Thank you very much for your constructive comment.
Nanoparticles are considered as a novel lubricant additive due to many advantages, such as different particle chemistries, nanosize to enter contact asperities and forming tribofilm without induction period [1-3]. Both EDS and XPS results show the MoS2 QDs existed on the wear scar. Besides, we also referred to some literature reports about the formation of MoS2 tribofilm on the sliding scar[4], meaning that the MoS2 tribofilm could occur with different concentration of MoS2 QDs, we think they show different compactness and tribological behaviours, but the tribo-film is existed. The relationship between forming process and the formation rate of MoS2 tribofilm is also an interesting topic, we will study in the next step.
[1] Bagi SD, Aswath PB. Role of MoS2 morphology on wear and friction under spectrum loading conditions. Lubrication Science 2015; 27:429–449.
[2] Xiaodong Z, Xun F, Huaqiang S, Zhengshui H. Lubricating properties of Cyanex 302‐modified MoS2 microspheres in base oil 500SN. Lubrication Science 2007; 19:71–79.
[3] Rosentsveig R, Gorodnev A, Feuerstein N, Friedman H, Zak A, Fleischer N et al. Fullerene‐like MoS2 nanoparticles and their tribological behavior. Tribology Letters 2009; 36:175–182.
[4] Wu H, Wang L, Dong G, et al. Lubrication effectiveness investigation on the friendly capped MoS2 nanoparticles[J]. Lubrication Science, 2017, 29(2): 115-129.
Reviewer 2 Report
In the attachment.

Reviewer 3 Report
There are a number of questions to the authors:
It remains unclear in what final form the quantum particles were obtained, in an aqueous solution or in a solid form? As carried out the dissolution of the nanoparticles in the base oil? What is the stability of nanoparticles in relation to oxygen, to the aggregation process?
There are a number of comments:
For figure 3 (d) on the ordinate axis, put the dimension (s); Most likely, there are errors in the figure caption 4, namely: 4 (b): EDS-result; 4(c): 0.1 Wt% MoS2 QDs; 4(d): 0.2 Wt% MoS2 QDs; 4(e): 0.3 Wt% MoS2 QDs; 4(f): 0.5 Wt% MoS2 QDs; The phrase on page 5 (lines 159-160) contradicts what was said above on page 4 (lines 128-130); It is necessary to specify errors in the determined values (COF, etc.)
Round 2
Reviewer 1 Report
Thank you for your explanations and changes in the manuscript.
In my opinion this manuscript in the presented form is acceptable for publication in Nanomaterials.